# Fish can use hydrostatic pressure to determine their absolute depth

Victoria A. Davis [1✉], Robert I. Holbrook[1] & Theresa Burt de Perera [1✉]

Hydrostatic pressure is a global cue that varies linearly with depth which could provide crucial spatial information for fish navigating vertically; however, whether fish can determine their depth using hydrostatic pressure has remained unknown. Here we show that Mexican tetras (*Astyanax mexicanus*) can learn the depth of a food site and consistently return to it with high fidelity using only hydrostatic pressure as a cue. Further, fish shifted their search location vertically as predicted if using pressure alone to signal depth. This study uncovers new sensory information available to fish which allows them to resolve their absolute depth on a fine scale.

[1] Department of Zoology, University of Oxford, Zoology Research and Administration Building, 11a Mansfield Road, Oxford OX1 3SZ, UK.
✉email: vdavis.research@gmail.com; theresa.burt@sjc.ox.ac.uk

For many animals traversing surfaces, the task of navigating is principally limited to the horizontal plane. In contrast, fish that move in three dimensions must also incorporate accurate vertical information into their representation of the environment if they are to navigate effectively[1,2]. Despite the necessity of this task, a reliable sensory cue that fish could use to achieve this is so far unknown. Whilst many fish have access to a variety of cues, most of these can become distorted or obstructed under certain environmental conditions, rendering them unreliable or misleading. For example, both light and acoustic cues rapidly attenuate with water depth, and visual cues can be obscured at night or in turbid conditions[3]. Unlike other sensory cues, hydrostatic pressure (hereafter referred to as pressure) is relatively stable over space and time[4] and is globally available throughout all bodies of water, rising predictably with increasing depth. Therefore, in principle, pressure is a reliable cue for resolving depth, and a sense of depth would provide a substantial adaptive advantage to fish that must navigate vertically in a complex and changeable environment[1]. But the question remains, can fish use pressure to determine their depth?

We know that fish are able to detect changes in pressure through the behavioural and physiological adaptations they use to maintain neutral buoyancy as they move through the water column[5–7]. However, this in itself is not an indication that fish can use pressure for navigation. Maintaining neutral buoyancy is a simple reflexive response governed by the autonomic nervous system[8], whereas effective navigation relies on more complex cognitive processes that require fish to learn and remember the spatial relationship between different locations[9] and use this information to establish the direction of their goal[10]. For fish to be capable of using pressure to navigate they must be able to ascribe depth information to their position in the water column and relate that information to other locations they may visit. Whether fish are capable of this has so far only been speculated[1,4].

Here we present empirical evidence that fish can determine the depth of a food site using only pressure as a navigational cue.

## Results

**Navigation under fixed pressure conditions**. Ten of the 27 fish from our sample consistently identified the location of the food site within 100 fixed pressure trials or fewer, despite visual cues being scrambled. This was a higher proportion of the group than we expected to find by chance (permutation test: $p < 0.001$). Why some individuals performed better than others at locating the food site in the fixed pressure condition is unclear. It is possible that these fish may have differed from the successful fish in noncognitive aspects, for example, how stressful they found the experimental task or how motivated they were to find the food[11–13], possibly preventing them from learning the location of the food site.

**Navigation under varied pressure conditions**. Fish adjusted their search depth according to the pressure change in 17 of the 30 trials (Fig. 1a,b; permutation test: $p < 0.001$), for example, choosing a flake 15 cm higher after we added 15 cm of water. We found no evidence of any bias in search depth with respect to the predicted depth since there was considerable overlap with 0 of our bootstrapped confidence intervals (95% C.I.: $-7.7 <$ estimate $< +2.2$). Further, we re-sampled, with replacement, our expected - observed (difference) data with a randomised sign (positive or negative) and found that, under this null scenario, a bias as big or greater than that observed in our data was seen on ~30% of 1,000 runs (Fig. 1a).

## Discussion

We have demonstrated that Mexican tetra fish can locate their depth with high fidelity by using hydrostatic pressure alone. Crucially, the fish can use hydrostatic pressure not only as a gradient, giving information about upward and downward movement but also as a distance-based cue that can allow precise localisation of their vertical position. This newly identified sensory capability indicates how fish can achieve the complex task of navigating through three-dimensional environments.

The basis of navigation in all animals, hinges on the individual knowing the spatial relationship between their current location and an intended destination. Although all animals inhabit a three-dimensional world, many, including humans, are constrained to travelling over surfaces with three degrees of freedom: two translational and one rotational[14]. The addition of the vertical dimension enlarges the size of the navigable space from a two-dimensional plane to a three-dimensional volume[2], leading to a multiplicative increase in the complexity of a navigational task[14–16]. Reliable information on vertical position would therefore be a significant benefit for three-dimensional navigation.

Although it is likely that in the wild fish rely on multiple cues to navigate, a sense of pressure would be particularly useful when other cues are unavailable or unreliable, for example, in turbid waters where visual landmarks are absent or obscured, and in turbulent waters where olfactory plumes cannot provide fine-scale information. The stability and ubiquity of hydrostatic pressure in aquatic environments allow fish access to a reliable navigational cue and could explain why two separate experiments, each testing a different species, found that fish perceived vertical information as the more reliable cue when horizontal and vertical information conflicted.

The physiological mechanism underlying depth perception in fish is yet to be identified, although the swim-bladder has been implicated. In this putative mechanism, absolute depth is estimated during fast, steady vertical displacements by combining a measurement of vertical speed with a measurement of the fractional rate of change of swim-bladder volume. If this is the mechanism that these and other bony fish are using to sense their depth, there are likely to be important ecological and welfare consequences for fish that suffer barotrauma from angling or transit through hydroelectric power facilities, where the damage caused from exposure to rapid changes in barometric pressure may cause swim bladder ruptures[17]. Therefore, governments need to be aware of key migratory paths that fish use to move between feeding and breeding sites to enable them to protect important species. Similarly, fish that contract parasitic infections of the swim bladder are likely to find their vertical navigation is severely compromised. While there are currently no studies on the pressure sensing in fish with parasitic infections of the swim bladder, previous research has reported that infected Koi carp (*Cyprinus carpio*) are less able to achieve and sustain neutral buoyancy and demonstrate abnormal swimming behaviour[18]. Similarly, silver eels (*Anguilla Anguilla*) infected with a swim bladder nematode experienced a loss of buoyancy resulting in them expending more energy while swimming, impeding their migration[19].

While the swim bladder appears to be a good candidate organ for sensing hydrostatic pressure in bony fish, many cartilaginous or deep-sea species do not possess a gas phase, despite still being able to navigate vertically. Previous research has suggested that instead of relying on fractional changes in swim bladder volume, these species may rely on the sensory afferents of their lateral line system; with evidence that swimming crabs (*Callinectes ornatus*), mud crabs (*Panopeus herbstii*) and dogfish (*Scyliorhinus canicula*) sense pressure changes via the bending of hair cells oriented to

sense either vertical or horizontal displacements[20]. The ability of fish to use hydrostatic pressure to accurately locate a point in the vertical dimension may be important for fisheries management. Known points of interest, for example, food sites, refugia, heavily predated areas, present in the vertical column could be learned and remembered by fish, with them either returning to or avoiding these areas as necessary. Further field studies on individual fish and shoals of fish using hydrostatic pressure in this context are needed to identify how this cue is used both in the wild and in farmed fisheries.

Our findings reveal novel sensory information that *A. mexicanus*, and possibly other fish species, use to gain detailed navigational information over short distances in the vertical dimension. Extrapolating from this, we argue that it is likely that fish could use pressure to navigate over larger distances as the pressure magnitudes will increase as the vertical distance increases. Together, this study reveals a new sensory capacity that has great adaptive value in the fish's volumetric world.

## Methods

**Apparatus**. We conducted all behavioural trials inside an experimental fish tank where we were able to manipulate the ambient pressure by adding or removing water from the tank, thus increasing, and decreasing the pressure, respectively, without influencing visual cues. To ensure fish could not use any visual cues inside the tank we installed a

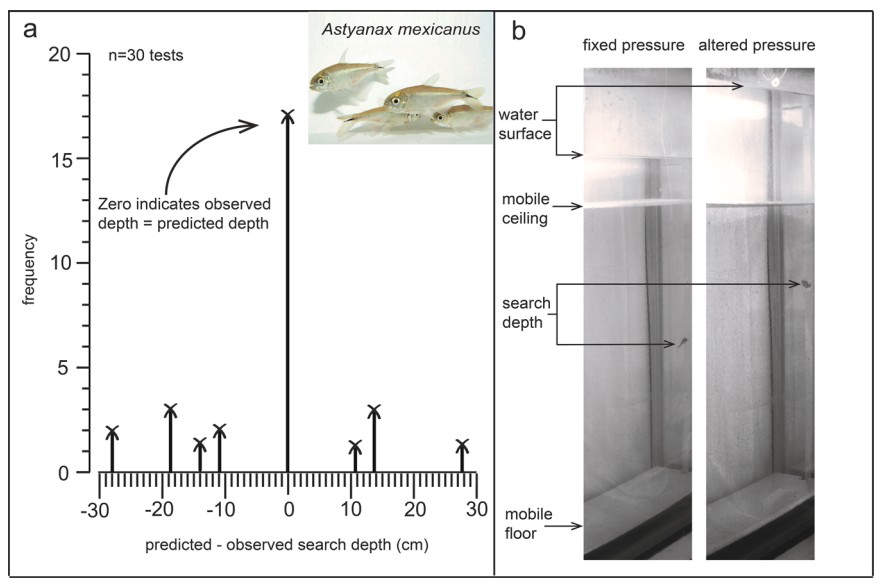

**Fig. 1 Difference in observed and predicted search depth in the altered pressure phase. a** Difference between the search depth we observed and the search depth we predicted according to the change in hydrostatic pressure in all 30 tests. A difference of zero means fish searched where we predicted according to hydrostatic pressure. **b** Stills from recorded footage of an individual during the fixed pressure phase (left) and the altered pressure phase (right) where we added 15 cm of water and observed the fish respond by searching 15 cm higher in the tank, as predicted.

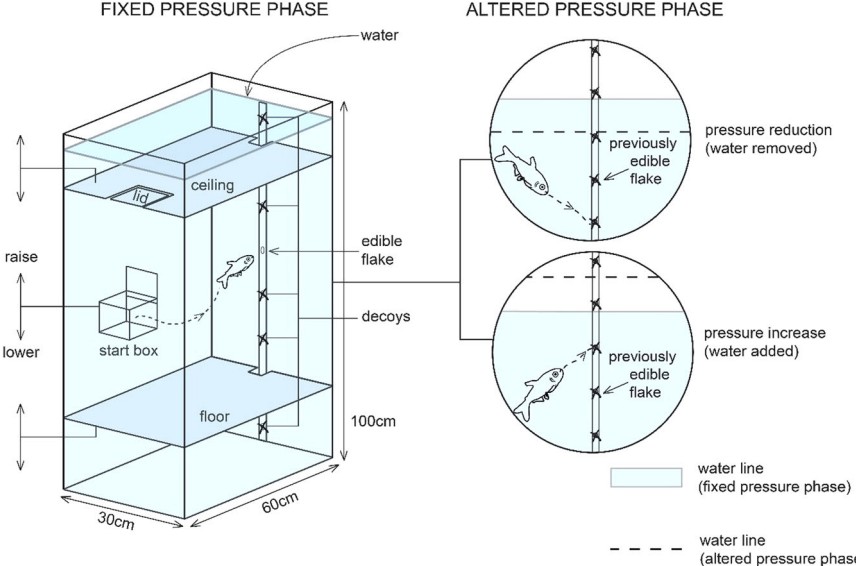

**Fig. 2 Schematic diagram of experimental setup during the fixed pressure phase and predicted search locations during the altered pressure phase.** Fixed pressure phase: The lower dark-grey platform represents the mobile floor and the upper platform represents the mobile ceiling. The small circle at the fish's mouth represents the edible flake fastened to a narrow rectangular piece of clear acrylic using petroleum jelly; the crosses on the acrylic represent the inaccessible decoy flakes. Altered pressure phase: diagram of predicted fish search depths during the altered pressure phase, representing the prediction after pressure was reduced (upper circle) and after pressure was increased (lower circle).

mobile floor and ceiling that we could raise or lower before each behavioural trial. This allowed us to change fish perceptions of the dimensions of the tank (distance from floor to ceiling) without influencing the pressure, preventing fish from using the upper and lower limits of the tank as landmarks for navigation.

### Experimental procedure

*Fixed pressure phase*. Our experiment was conducted in two phases: a *fixed pressure* phase and an *altered pressure* phase. During the fixed pressure phase, we did not manipulate the hydrostatic pressure in the tank, instead we tested whether fish could remember the location of a piece of food under constant pressure conditions. We presented fish with a single edible food flake (positioned at the same depth during every trial) amid a vertical line of between three and six decoy flakes – these looked identical to the edible flake except that they were protected by glass coverslips so could not be consumed or detected by olfaction (Fig. 2 and Supplementary Fig. 1). We varied the distance between the flakes presented during each trial so that fish could not 'count' or use visual patterns to discern the edible flake among the decoys (for the different combinations of experimental conditions, see Supplementary Figure 2). We also adjusted the depth from which fish were released at the beginning of each trial to prevent them from memorising a trajectory to reach the edible flake.

*Altered pressure phase*. Those fish that were successful at locating the food on their first attempt in the fixed pressure phase were then tested under the altered pressure phase conditions. Here, we tested whether fish would alter the depth they searched for the food after a change in pressure (Supplementary Fig. 3). During the altered pressure phase all flakes were decoys, this was to rule out the possibility that fish could use olfactory cues (that would have emanated from an uncovered flake) as a navigational cue. Fish were either presented with an increase in pressure where we added water to the tank ($n = 4$) or a decrease in pressure where we reduced the water in the tank ($n = 6$). Space constraints inside the tank meant that we could not test fish under both conditions (see Supplementary Methods for more details). Each fish ($n = 10$) received three trials during this phase ($n = 30$).

### Data analysis

To compare fish search depths between the fixed and altered pressure phases we paired each of the 30 trials in the altered pressure phase to trials in the fixed pressure phase that were matched for ceiling and floor height, and flake number. We then 'drew' a flake at random chosen by each fish during one of these trials during the fixed pressure phase and treated this as the flake chosen in the altered pressure phase. Finally, we permuted these 'choices' 1000 times and counted how often the results matched what we observed. To test whether the error distribution of fishes' search depths was skewed, which would indicate a bias in their search depth—either more shallow or more deep in the water column than expected by chance, we bootstrapped the mean difference to obtain confidence intervals.

### Statistics and reproducibility

Statistical tests were performed in R version 3.3.3. The fish's ability to learn and remember the depth of the food site (fixed pressure phase) was tested by comparing their real choices to random choices where there was an equal probability of them choosing any flake. To assess the fish's ability to respond to a pressure change we permuted fish's real choices in the fixed pressure phase and compared them to their choices in the altered pressure phase. The criterion for statistical significance was $P < 0.05$ for all tests.

**Reporting Summary**. Further information on research design is available in the Nature Research Reporting Summary linked to this article.

## Data availability

Data collected and analysed during this study can be found in the supplementary materials. Source data used to generate Fig. 2a is supplied as Supplementary Data 1.

## Code availability

All custom code deemed central to the conclusions can be obtained from the corresponding authors upon reasonable request.

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

## Acknowledgements
We thank Callum Lawson, Oliver Padget, Graham Taylor and Alan Grafen for their advice about the statistical analysis. This work was supported by a Biotechnology and Biological Sciences Research Council Doctoral Training Grant (BB/H01103X/1), VAD also received support from St. John's College, University of Oxford.

## Author contributions
R.I.H., T.B.d.P. and V.A.D. designed the experiment. V.A.D. carried out the behavioural trials and wrote the article. All authors discussed the results and commented on the manuscript.

## Competing interests
The authors declare no competing interests.

## Ethical compliance
Ethical approval for this study was given by Zoology AWERB (Animal Welfare and Ethics Review Board). This study did not require a Home Office License.
