## [Peer Review File · Communications Biology]

Reviewers' comments:

Reviewer #1 (Remarks to the Author):

The authors have submitted a novel and well written article describing the methods and results of testing to determine if fish can detect hydrostatic pressure and use that sense to assist in navigation within three-dimensions. I personally find this article very interesting and applicable to my own research. The methods and results are well described and support the conclusion. I believe this article should be accepted for publication with some minor revisions.

Though I appreciate that the discussion is succinct and to the point, I feel that it is lacking the implications and/or applications of these findings and should be briefly expanded. In expanding the discussion, the authors may want to consider some of the following questions:

- Could this new knowledge be applied to fisheries management?
- What about mitigating man-made stressors on fish? For example, if the mechanism is attributed to the swim bladder, what does that mean for fish that suffer barotrauma to the swim bladder from angling or passage through hydropower facilities? Would this ability be impaired?

Additionally, the discussion about the mechanism of this sense is very brief and could be expanded. For example, some questions came to my mind when reading this:

- What about the lateral line, could that also be used to sense depth?
- And how about fish without swim bladders such as lamprey, sharks, and flat fish, would they not have this ability? Potentially more evidence to support use of the lateral line.
- Would the presences of swim bladder diseases or parasites affect this ability, or as mentioned before swim bladder injuries due to decompression (angling, turbine passage)?

Specific Comments:

Line 62: it would be helpful here to mention if the glass cover slips also prohibit or reduce the fish's ability to locate the decoys through olfactory sensors.

Line 102: are the training trials what were conducted during the fixed pressure phase, or is this different?

Line 128: the abstract and the supplementary data state that Mexican tetras were used, but here it states banded tetra.

Reviewer #2 (Remarks to the Author):

This work addresses the question of hydrostatic pressure sensibility in fish Mexican Tetras. The authors use two different experiments (fixed and altered pressure, respectively), to show empirically whether fish could determine the depth of a food site using pressure as a navigational cue.

I find the question interesting and, as mentioned by the authors, the problem of pressure sensibility to determine the depth is still to be elucidated.

However, I find the study too empirical. If the two setups shown are nice, the data lack robustness. I also believe that the sample is not statistically to give strong conclusions.

I think that this work needs to be expended before any publication in Communication. Biology.

Reviewer #3 (Remarks to the Author):

This is a well-constructed study showing fish can determine the depth of a food site using pressure as a cue.

I have a few suggestions for the authors consideration:

1. The authors' own summary of the study is that they have "tested empirically whether fish could determine the depth of a food site using pressure as a navigational cue." My question is: does orientation over this scale warrant the use of the term navigation?

2. I haven't accessed the supplementary material so this may be answered in there, but it would be useful to have a time scale on the "altered pressure phase" experiments. Presumably there is a relatively short time delay between the 1st choice/pressure change/2nd choice. If this is the case, then it makes sense that pressure changes measured by the swim-bladder could provide the appropriate cue. However, it would be the case that over longer time scales, swim-bladder adjustments (inflation/deflation to maintain buoyancy) would mean that the swim-bladder wouldn't provide an absolute sense of hydrostatic pressure.

3. Which brings me to the 3rd point. There is some evidence for absolute pressure sensing in dogfish (in the absence of a swim-bladder: [Fraser, P. J., Cruickshank, S. F., Shelmerdine, R. L., & Smith, L. E. (2008). Hydrostatic pressure receptors and depth usage in Crustacea and fish. *Navigation*, 55(2), 159-165.) It may be possible to address the swim-bladder/other sensory origins for hydrostatic sensing using an extension to the methods used in this study. E.g. if the fish is held for a range of periods after the pressure change, then one might predict a systematic drift in the error of choice in the direction expected from the swim-bladder adjustment to the pressure change.

Fish can use hydrostatic pressure to determine their absolute depth – resubmission to
Communications Biology

Department of Zoology
University of Oxford
Oxford, OX1 3SZ

19/08/2021

Dear Reviewers,

We thank you for your positive feedback and detailed comments regarding our manuscript entitled “Fish can use hydrostatic pressure to determine their absolute depth” [COMMSBIO-21-0992-T]. You have highlighted important details, particularly areas we could expand on, and we feel these comments have greatly improved the quality of this manuscript. Our responses to your comments have been detailed below. Reviewers’ original comments are written in plain font on the left side of the table, our responses and the actions we have taken follow on the right side of the table, and any text that we lift directly from the manuscript is in quotations and italicised. All changes have been highlighted by the line number and correspond to the tracked version of the revised manuscript.

Yours, sincerely,

Victoria A. Davis

On behalf of Robert I. Holbrook and Theresa Burt de Perera

Detailed response to individual author comments:

Line number	Reviewer’s Comment	Authors’ Response
Reviewer #1		
N/A	Though I appreciate that the discussion is succinct and to the point, I feel that it is lacking the implications and/or applications of these findings and should be briefly expanded. In expanding the discussion, the authors may want to consider some of the following questions:  • Could this new knowledge be applied to fisheries management? 	We thank the reviewer for their suggestion and have added the following text to the manuscript (lines 173-179): “The ability of fish to use hydrostatic pressure to accurately locate a point in the vertical dimension may be important for fisheries management. Known points of interest, for example, food sites, refugia, heavily predated areas, present in the vertical column could be learned and remembered by fish, with them either returning to or avoiding these areas as necessary. Further field studies on individual fish and shoals of fish using hydrostatic pressure in this context are needed to identify how this cue is used both in the wild and in farmed fisheries.”
N/A	 • What about mitigating man-made stressors on fish? For example, if the mechanism is attributed to the swim bladder, what does that mean for fish that suffer barotrauma to the swim bladder from angling or passage through hydropower facilities? Would this ability be impaired? 	We have expanded on this idea in the discussion (lines 152-158): “If this is the mechanism that these and other bony fish are using to sense their depth, there are likely to be important ecological and welfare consequences for fish that suffer barotrauma from angling or transit through hydroelectric power facilities, where damage caused from exposure to rapid changes in barometric pressure may cause swim bladder ruptures [17]. Therefore, governments need to be aware of key migratory paths that fish use to move between

Fish can use hydrostatic pressure to determine their absolute depth – resubmission to Communications Biology

		feeding and breeding sites to enable them to protect important species.”
N/A	Additionally, the discussion about the mechanism of this sense is very brief and could be expanded. For example, some questions came to my mind when reading this:  • What about the lateral line, could that also be used to sense depth? • And how about fish without swim bladders such as lamprey, sharks, and flat fish, would they not have this ability? Potentially more evidence to support use of the lateral line. 	We have addressed this by discussing species that do not possess a swim bladder the first two parts of this comment, adding the following text (lines 166-172): “While the swim bladder appears to be a good candidate organ for sensing hydrostatic pressure in bony fish, many cartilaginous or deep-sea species do not possess a gas phase, despite still being able to navigate vertically. Previous research has suggested that instead of relying on fractional changes in swim bladder volume, these species may rely on the sensory afferents of their lateral line system; with evidence that swimming crabs (Callinectes ornatus), mud crabs (Panopeus herbstii) and dogfish (Scyliorhinus canicula) sense pressure changes via the bending of hair cells oriented to sense either vertical or horizontal displacements [20].”
	 • Would the presences of swim bladder diseases or parasites affect this ability, or as mentioned before swim bladder injuries due to decompression (angling, turbine passage)? 	We have added the following passage addressing the potential effects of swim bladder parasite infestation (lines 158-165): “Similarly, fish that contract parasitic infections of the swim bladder are likely to find their vertical navigation is severely compromised. While there are currently no studies on pressure sensing in fish with parasitic infections of the swim bladder, previous research has reported that infected Koi carp (Cyprinus carpio) are less able to achieve and sustain neutral buoyancy and demonstrate abnormal swimming behaviour [18]. Similarly, silver eels (Anguilla Anguilla) infected with a swim bladder nematode experienced a loss of

Fish can use hydrostatic pressure to determine their absolute depth – resubmission to Communications Biology

		buoyancy resulting in them expending more energy while swimming, thus impeding their migration [19].”
62	It would be helpful here to mention if the glass cover slips also prohibit or reduce the fish’s ability to locate the decoys through olfactory sensors.	We have added to the original sentence to confirm this (line 62): “... or detected by olfaction (Figure 1).”
102	Are the training trials what were conducted during the fixed pressure phase, or is this different?	That is correct, however, to remain consistent and prevent confusion we have replaced “training trials” with “fixed pressure” trials.
128	The abstract and the supplementary data state that Mexican tetras were used, but here it states banded tetra.	The reviewer is correct, this was a mistype, the species used were the Mexican tetra. This has now been amended in the text.
Reviewer #2		
N/A	I find the question interesting and, as mentioned by the authors, the problem of pressure sensibility to determine the depth is still to be elucidated. However, I find the study too empirical. If the two setups shown are nice, the data lack robustness. I also believe that the sample is not statistically to give strong conclusions. I think that this work needs to be expended before any publication in Communications Biology.	We are pleased that the reviewer believes that we are addressing an interesting question. We are unsure what the reviewer means by an empirical study being “too empirical”. The reviewer also appears to agree that the setups of the experiment are “nice”, but they have concerns that our data lack robustness. It is unclear how making the study less empirical would increase the robustness of the statistics. Further, the reviewer does not state how or why they consider the data to not be robust. It is further unclear what the reviewer means by “robustness” in this context. We present statistical analyses using the sample size we managed to get, which reveal significance at that sample size. Further, while this may not matter from a strictly scientific point of view, the equipment and indeed the building used to conduct the experiments in have since been destroyed. It is therefore not possible to conduct further experiments to increase the sample size.

Fish can use hydrostatic pressure to determine their absolute depth – resubmission to Communications Biology

Reviewer #3		
N/A	The authors' own summary of the study is that they have “tested empirically whether fish could determine the depth of a food site using pressure as a navigational cue.” My question is: does orientation over this scale warrant the use of the term navigation?	We appreciate the reviewer’s concerns, however we feel that navigation is more appropriate in this context despite it being over such a short distance, as orienting is broadly understood as aligning oneself relative to a particular compass direction, however, we were interested in both the fish’s angular alignment and the vertical distance travelled.
N/A	I haven’t accessed the supplementary material so this may be answered in there, but it would be useful to have a time scale on the “altered pressure phase” experiments. Presumably there is a relatively short time delay between the 1st choice/pressure change/2nd choice. If this is the case, then it makes sense that pressure changes measured by the swim-bladder could provide the appropriate cue. However, it would be the case that over longer time scales, swim-bladder adjustments (inflation/deflation to maintain buoyancy) would mean that the swim-bladder wouldn’t provide an absolute sense of hydrostatic pressure.	This is correct; however, we had not previously included these details in the supplementary material, this has now been added (lines 129-134 of supplementary material). “Altered pressure tests were performed immediately after fixed pressure tests were over to ensure that the swim bladder was primed to detect the pressure change, as over longer time scales, the swim-bladder would have adjusted (either by inflating or deflating) to allow the fish to maintain neutral buoyancy, and thus prevented fish from deriving an absolute sense of hydrostatic pressure from the swim bladder.”
	Which brings me to the 3rd point. There is some evidence for absolute pressure sensing in dogfish (in the absence of a swim-bladder: [Fraser, P. J., Cruickshank, S. F., Shelmerdine, R. L., & Smith, L. E. (2008). Hydrostatic pressure receptors and depth usage in Crustacea and fish. Navigation , 55(2), 159-165.) It may be possible to address the swim-bladder/other sensory origins for hydrostatic sensing using an extension to the methods used in this study. E.g. if the fish is held for a range of periods after the pressure change, then one might predict a systematic drift in the error of choice in the	We thank the reviewer for their suggestion of how to extend the methods. We agree that it would be really interesting to further interrogate the site of pressure sensing however, this was beyond the scope of this project. We thank the reviewer for the information on likely mechanisms for absolute depth perception in other species, and we have now cited the Fraser et al. paper in our manuscript.

Fish can use hydrostatic pressure to determine their absolute depth – resubmission to Communications Biology

	direction expected from the swim-bladder adjustment to the pressure change.	
--	---	--